# Wearable Focal Muscle Vibration on Pain, Balance, Mobility, and Sensation in Individuals with Diabetic Peripheral Neuropathy: A Pilot Study

**DOI:** 10.3390/ijerph18052415

**Published:** 2021-03-02

**Authors:** Raghuveer Chandrashekhar, Hongwu Wang, Carol Dionne, Shirley James, Jenni Burzycki

**Affiliations:** 1Department of Rehabilitation Sciences, College of Allied Health, University of Oklahoma Health Sciences Center, Oklahoma City, OK 73117, USA; Raghuveer-Chandrashekhar@ouhsc.edu (R.C.); Carol-Dionne@ouhsc.edu (C.D.); Shirley-James@ouhsc.edu (S.J.); Jenni-Burzycki@ouhsc.edu (J.B.); 2Peggy and Charles Stephenson School of Biomedical Engineering, University of Oklahoma, Norman, OK 73019, USA

**Keywords:** wearable focal muscle vibration, diabetic peripheral neuropathy, pain, balance and mobility, sensation, satisfaction and compliance

## Abstract

People with diabetic peripheral neuropathy (DPN) experience a lower quality of life caused by associated pain, loss of sensation and mobility impairment. Current standard care for DPN is limited and lacking. This study explores the benefits of 4-week, in-home wearable focal muscle vibration (FMV) therapy on pain, balance, mobility, and sensation in people with DPN. Participants were randomized into three groups and received different FMV intensities. FMV was applied using a modified Myovolt^TM^ wearable device to the tibialis anterior, distal quadriceps, and gastrocnemius/soleus muscles on both lower limbs for three days a week over four weeks. The outcomes included pain, balance, mobility, sensation, device usage log, feedback survey, and a semi-structured interview. In all, 23 participants completed the study. The results showed significant improvement in average pain (Pre: 4.00 ± 2.29; Post: 3.18 ± 2.26; *p* = 0.007), pain interference with walking ability (Pre: 4.14 ± 3.20; Post: 3.09 ± 1.976; *p* = 0.03), and standard and cognitive Timed Up-and-Go scores (Pre: 13.75 ± 5.34; Post: 12.65 ± 5.25; *p* = 0.04; Pre: 15.12 ± 6.60; Post: 12.71 ± 5.57; *p* = 0.003, respectively); the overall pain improvement was trending towards significance (Pre: 3.48 ± 2.56; Post: 2.87 ± 1.85; *p* = 0.051). Balance and sensations improved but not significantly. There was a trend towards significance (*p* = 0.088), correlation (r = 0.382) between changes in balance and baseline pain. The participants were highly satisfied with wearable FMV and were 100% compliant. FMV therapy was associated with improved pain, mobility, and sensation. Further study with a larger sample and better outcome measures are warranted.

## 1. Introduction

Unless an effective intervention is found for diabetic peripheral neuropathy (DPN), one third of the 9.7 billion people in the world with diabetes will suffer its effects by 2050 [1]. DPN affects approximately 50–70% of patients with diabetes and is the most common debilitating secondary complication [2,3]. DPN is characterized by loss of sensation in the lower extremities in a distal to proximal progression. Specifically, patients with DPN experience pain, the lack of proprioception, and loss of muscle strength (usually in the toe and ankle extensors), poor kinesthetic sense, and a lack of vibratory sense [4,5,6]. The combined effects of these symptoms cause impairment in postural stability, balance, and functional mobility. This affects each patient’s ability to ambulate safely and independently and lowers their quality of life (QOL). Additionally, DPN significantly increases healthcare costs associated with diabetes. In 2012, the total annual health care cost associated with diabetes in the U.S. was $245 billion, 27% of which was attributable to DPN [3]. An effective and cost-effective intervention is imperative.

Balance and mobility are integral for safe activities of daily living (ADL). Balance, defined as the ability to maintain upright posture, is composed of static and dynamic components [7]. While static balance is the ability to hold a position, dynamic balance is the ability to maintain stability while transitioning between positions. Postural control, comprised of both static and dynamic balance, is significantly impaired in individuals with DPN. Poor balance is most evident in the medial-lateral (frontal) plane. Researchers have demonstrated an association between maximum medial-lateral dynamic sway and the extent of peripheral neuropathy [7]. Shumway-Cook and Woollacott [8] note that balance deficits are the strongest predictor of falls, especially during more complex activities [8]. In addition, individuals with DPN experience a fear of falls, which leads to a more sedentary lifestyle that increases the overall progression of diabetes-linked nerve damage, thereby exacerbating the symptoms of DPN [9].

While researchers have studied the effects of various pharmacological and non-pharmacological interventions such as duloxetine, supplements (alpha-lipoic acid), pregabalin, spinal cord stimulation, transcutaneous electrical stimulation (TENS), and whole body vibration (WBV), used to address the symptoms experienced by individuals with DPN [2], there is a lack of consensus about the success of these interventions. Of these, randomized controlled trials (RCTs) [10,11,12] that examine the effect of WBV in individuals with DPN show a significant association with decreased pain, improved balance, and heighted gait performance. WBV is a form of mechanical stimulation shown to reduce acute pain, improve balance and dynamic stability, improve glycemic control, and increase muscle strength in individuals with DPN [13,14]. While WBV is an effective intervention for DPN, it is also associated with tissue inflammation and potential adverse effects on the nervous and vascular tissues [15]. Unfortunately, WBV devices currently available significantly exceed the International Organization for Standardization (ISO) guidelines for safety [16]. However, researchers studying vibration as a rehabilitation intervention shown evidence that focal vibration benefits spasticity, motor learning deficits, pain, balance and mobility impairment in patients with stroke, spinal cord injury, and multiple sclerosis [17].

Focal vibration or focal muscle vibration (FMV), an innovative form of vibration, is a non-invasive intervention that applies a mechanical stimulus to specific muscles, tendons, or regions of choice [17]. Unlike WBV, its application can easily be kept within safe limits [17]. FMV promotes neural plasticity and long-lasting motor recovery [17]. When applied repeatedly, FMV produces a repeated sensory input that reaches the primary motor cortex (M1) directly via Ia fiber afferent input, thereby leading to an improvement in motor function by means of an intrinsic plasticity-related mechanism [17,18,19,20]. Recent literature has established evidence supporting the benefits of FMV in activating the primary somatosensory cortex and intensifying the connection strength of the central region [17,19,21], increasing nitric oxide production [22], improving blood flow [23], and increasing angiogenesis [24]. 

The application of FMV has yet to be investigated in individuals with DPN but is potentially beneficial in addressing painful symptoms of neuropathic pain as well as walking impairments. In addition, FMV delivered in a wearable format could be applied at home and community settings and be used while patients performing functional activities. The primary aim of this exploratory study was to explore the benefits over 4 weeks of wearable FMV therapy on pain, balance, mobility, and sensation in individuals with DPN. The secondary aim was to determine if the intensity of vibration had varying effects on changes in balance, mobility, pain or sensation. Finally, this study assessed whether pain presented at baseline was associated with changes associated with the aforementioned effect of FMV.

## 2. Materials and Methods

### 2.1. Study Design

We conducted a pilot feasibility study with a single-blind, parallel-group randomized design in the Technology for Occupational Performance Laboratory at the University of Oklahoma Health Sciences Center (OUHSC). The study was approved by the OUHSC Institutional Review Board (#9688). 

### 2.2. Subjects

Participants were recruited and randomly assigned to one of three intervention groups after they gave their consent: Intervention group 1 received a pulsed on-and-off vibration at 120 Hz; intervention group 2 received sinusoidal vibration ranging between 35 and 120 Hz; and intervention group 3 received continuous vibration at a constant frequency of 120 Hz. Randomization codes were generated prior to the study for a 1:1:1 allocation ratio and stored in an Excel spreadsheet. The randomization was then completed using permutated blocks of 3 or 6. At the time of this study, we were not able to provide sham vibration with the wearable device, and when we tried to allocate the participant to the control group, the participant refused to participate, especially after an experience with the FMV. Therefore, we were not able to include a control group in this study. Inclusion criteria for this study included a diagnosis of Type II Diabetes Mellitus, a secondary diagnosis of DPN of a one-year duration, aged 18 years and older, the ability to ambulate independently, English speaking, and normal or corrected vision. To ambulate independently is defined as the ability to ambulate without supervision or physical assistance from another person. Assistive devices, orthoses, and prostheses are allowed. Exclusion criteria included neuropathy not related to DM, symptomatic peripheral vascular disease, joint pain, swelling and a limited range of motion in the lower extremities that interferes with walking, lower extremity amputation, and scores of less than 24 on the Montreal Cognitive Assessment (MOCA) [25]. 

#### Sample Size

Because no previous research had been conducted on the effect of focal vibration in patients with DPN, we performed power calculations using minimal clinically important differences (MCID) when this information was available, as well as reasonable estimates of expected change. We based our calculations on our primary outcomes: TUG, pain and BBS. With an alpha = 0.05, a power of 0.8, and using a paired t-test, we needed eight subjects per group to detect within subject changes.

### 2.3. Outcome Measures

The outcome measures used in this study were the Berg Balance Scale (BBS) [26], the Cognitive and standard Timed-Up and Go (TUG) [27], the Semmes–Weinstein Monofilament Test (SWMT) with the 5.07 (10 g) filament [28], and the Brief Pain Inventory—Diabetic Peripheral Neuropathy (BPI–DPN) [29]. These measures were selected based on previous use and on established reliability and validity in patients with DPN. The device feedback survey was conducted using the assistive technology subscale of the Quebec User Evaluation of Satisfaction with Assistive Technology questionnaire (QUEST 2.0) [30]. The assistive technology subscale of the QUEST 2.0 consists of 8 items with a total possible score of 40, where each item is scored on a scale of one to five [30].

### 2.4. Intervention

FMV was applied using a modified version of a commercially available wearable focal vibration device (Myovolt^TM^, Myovolt Limited, Christchurch, New Zealand) (Figure 1a). Myovolt™ is registered with the U.S. Food & Drug Administration (Regulation Number: 890.5660) under the “therapeutic massager” category and has been used as a muscle stimulation device for sports massage in a similar way. Studies using this device for athletes reported the positive effects of increasing peripheral blood circulation, and reducing muscle soreness [31,32]. In this study, each of the three intervention groups received a different vibration intensity. Intervention group 1 received a pulsed on-and-off vibration at 120 Hz; intervention group 2 received sinusoidal vibration ranging between 35 and 120 Hz; and intervention group 3 received continuous vibration at a constant frequency of 120 Hz. The selection of those three vibration intensities was based on previous research studies on FMV-applied frequencies, the majority of which ranged from 30 Hz to 120 Hz [17], and previous studies had shown those three intensities to be safe [31,32]. Additionally, the wearable FMV device available at the time of the study had only those three vibration intensities. We hypothesized that patients with DPN would respond differently to different intensities due to their sensory loss.

### 2.5. Procedures

During intervention, patients wore the modified Myovolt™ vibration device on both legs, applying FMV to the tibialis anterior, the distal quadriceps, and the belly of the gastrocnemius/soleus muscle (Figure 1b). The original Myovolt device came with two vibration motors for each unit, and the delivery of each motor was affected by another one. The original device also allows the user to adjust the vibration intensity among the three intensities mentioned above. To better quantify the vibration delivered and control what vibration the participant received, in this pilot study we asked the manufacture to modify the device with one vibration motor for each unit. Each muscle was vibrated for 10 min (total 30 min for each leg), with an intersession interval of one minute per day, three days a week, for four weeks. A vibration intensity of 35–120 Hz, was used in previous studies [17]. Participants were trained to operate, attach, and detach the device. Investigators collected demographic information on each patient’s first visit. All participants also completed a log of their usage of the device during the intervention and a device feedback survey with a semi-structured interview during their second visit.

### 2.6. Data Analysis

Descriptive statistics were reported for demographic measures, primary outcome measures, and the device feedback survey. A two-by-three, mixed-method analysis of variance (ANOVA)––pre-survey and post-survey x the three vibration groups––was computed to determine the effect within the subject before and after the vibration, the effect of the vibration group among subjects, and the interaction effect between the time and vibration group on each outcome measure. Like the gait study, differences on the outcome measures among individuals who use mobility related assistive technology (AT) and those who do not use AT devices were examined using a two-way mixed ANOVA. It was assumed that residuals both within subjects and between subjects would be normally distributed. Homogeneity of variances, or homoscedasticity, using the Box’s M statistic, and the Mauchly’s test of sphericity for the ANOVA tests were examined and used to report the appropriate ANOVA results. Pearson correlation tests or Spearman’s rho were used, respectively, to examine whether baseline pain data had an impact on changes in balance and mobility. SPSS 25 (IBM Corporation, Armonk, NY, USA) was used with an alpha level of 0.05 for each analysis. The interview data were analyzed to identify themes.

## 3. Results

The demographic data for all participants is summarized in Table 1. Of the 24 participants recruited, 23 completed both visits, and 21 had both baseline and post-intervention scores for all the primary outcomes. One participant did not complete the post-intervention TUG cognitive and another only completed the sensation and pain questionnaires at baseline as well as post-intervention.

Apart from the BBS scores and SWMT scores for the right foot, the scores of all other outcome measures at baseline and post-intervention were normally distributed (Kolomogorov–Smirnov test; *p* > 0.05). The TUG (Mean ± SD difference: 2.4117 ± 3.3160; *p* = 0.035, N = 22), and TUG cognitive (Mean ± SD difference: 1.1048 ± 2.2971; *p* = 0.003, N = 21) showed statistically significant improvement (*p* < 0.05) post-intervention (Table 2). While the BPI–DPN scores were trending toward significance (*p* = 0.051, N = 22), the BBS (*p* = 0.112, N = 22), the SWMT scores for the right foot (*p* = 0.428, N = 23) and left foot (*p* = 0.108, N = 23) were not statistically significant (Table 2). Although improvement in the overall pain score was trending towards statistical significance, a closer look at the subscales of the BPI–DPN (Table 3) revealed that FMV improved average pain (*p* = 0.007) and pain interference with walking ability (*p* = 0.024). The ANOVA did not reveal statistically significant differences with regard to vibration groups (Table 4). When examining the interaction effect as shown in Figure 2, though not significant, Groups 1 and 2 showed larger changes in pain as well as left and right foot sensation scores after the intervention. Groups 2 and 3 showed greater improvements in the BBS and TUG scores. Compared to the non-AT group, the AT group had significantly worse balance (*p* = 0.007) and mobility (*p* = 0.004 and 0.011 for TUG and TUG cognitive) at baseline as shown in Table 5. No significant interaction effect between time and usage of AT was observed, but the AT group showed clinically significant improvement on BBS (an 8-point increase) while there were almost no changes in the non-AT group. The improvements in pain and sensation were about the same between the two groups. The non-AT group showed overall better improvement on TUG and TUG cognitive than the AT group (Table 5). The changes in BBS were not normally distributed, so Spearman’s rho test was used for baseline pain and BBS. A trend towards significance (*p* = 0.088) as well as a positive correlation (r = 0.382) was observed between baseline pain measures and changes in BBS. However, no significant correlation was found between baseline pain scores and changes in TUG and TUG-cognitive scores.

Of the 23 participants who completed both visits, one participant failed to complete the feedback survey and interview. Among the remaining 22, 59% (13 out of 22) scored 35 and above out of a possible 40 in the QUEST 2.0 satisfaction survey with an overall mean ± SD satisfaction score of 34.41 ± 6.03. Device usage, perceived benefits of vibration therapy, and comments on wearable focal vibration were identified as the three most common themes observed in the results from the feedback survey and interview. Participant compliance was calculated as the ratio of number of sessions or days the participants used the FMV device to the recommended number of sessions. Two out of the 23 participants did not provide their device use log. Of the 21 participants who logged the device usage, the compliance rate was 100%. A post-hoc power analysis was conducted on the BPI–DPN, TUG, and TUG cognitive scores as they were the three outcomes with a moderate effect size greater than 0.4. This post-hoc analysis revealed that, based on the BPI–DPN scores (effect size = 0.45), TUG scores (effect size = 0.49), and TUG cognitive scores (effect size = 0.73), the study was powered at 52%, 59%, and 88%, respectively.

## 4. Discussion

To the investigators’ knowledge, this study was the first to explore the benefits of in-home, wearable FMV intervention for DPN symptom management. The preliminary results found significant improvement in the mobility as assessed by the TUG, TUG cognitive, and the average pain and pain interference to walking ability subscales of the BPI–DPN, as well as a trend towards significance in the overall pain scores. At baseline, 23 participants had an average BBS score of 42.14 points, which is comparable to the average BBS score of 43.7 in individuals without DPN as reported by Timar et al. [34]. In the current study, the lack of improvement in balance can be explained by the large variation in the baseline BBS scores (Table 2) as well as the relatively high baseline BBS scores, that is, the ceiling effect: 13 out of 23 participants had BBS scores greater than 45, which is the cut-off score used to determine fall risk. Although a statistically significant improvement in balance was not observed, a closer look at the baseline and post-intervention scores indicated a larger improvement in BBS scores in individuals with lower baseline BBS scores, suggesting that FMV may work better for those individuals with poorer balance at baseline.

The significant improvement in the TUG scores and TUG cognitive scores indicates an improvement in functional mobility, which is further supported by the significant improvement in the “pain interference with walking ability” subscale of the BPI–DPN. The trend towards significance in the overall pain score is corroborated by the improvement in the average pain subscale of the BPI–DPN, which further demonstrates that FMV could reduce pain. This agreed with the interview feedback obtained from the participants, stating that the intervention was quite enjoyable, and overall they experienced less pain and greater comfort while using the Myovolt device. Future studies with larger sample sizes could further validate this finding. The results observed in this study were comparable to similar studies conducted on individuals with DPN using exercise or WBV as intervention strategies [10,11,12,35,36,37,38]. Trends of improvement observed in balance, pain levels, and functional mobility––among other symptoms in studies that use exercise or WBV as the intervention for individuals with DPN––are comparable to the results obtained in this study. For instance, Lee et al. [12] reported an improvement in TUG score with a mean difference of 1.79 s [12], which is comparable to the TUG score mean difference of 1.10 s obtained in this study. Similarly, in studies examining the effects of exercise, the reduction in pain levels measured by the visual analog scale (VAS) and neuropathic pain scale (NPS) revealed an approximate decrease of six points out of a possible 10 [10] in individuals with DPN who had baseline pain levels greater than 5, which is consistent with the decrease in pain observed in our study. Individuals with higher pain levels had larger reductions in pain. The findings in this study suggest that the effects of FMV are comparable to that of WBV and exercise. The sensation was not significantly improved, but a larger improvement was observed in the left foot over the right foot. We observed improvements of sensation for those participants with lower sensation scores at baseline, and this is corroborated by the same participants reporting better sensation during the interview. Non-significant changes in sensation could be due to the small sample size, large variations in sensation scores at baseline, under-dosage of the vibration (several participants commented not receiving enough vibration), or the usage of the SWMT with only the 5.07 (10 g) filament, which might not have been sensitive enough to detect smaller changes in sensation.

Pain caused by DPN is a significant complication of diabetes and affects a patient’s functioning and well-being. In this study, we found that the change in balance after intervention was positively correlated with baseline pain scores, indicating that the worse the pain at baseline, the greater the improvement in the participant’s balance. This finding is in agreement with the results of a previous study, which showed that DPN pain had a negative impact on balance and mobility [39]. Because our FMV intervention reduced pain, even though the changes in balance were not statistically significant, patients who experienced more pain at baseline tended to have larger improvements in their balance after the intervention.

The lack of significance between group differences discovered from the ANOVA can be attributed to the small samples within each group. After intervention, we observed that the changes in pain and sensation were larger in group 1 with pulsing vibration and group 2 with sinusoidal vibration than in group 3 with continuous vibration as shown in Figure 2 and Table 4. As patients with DPN experience sensory loss, particularly vibratory sensation, we hypothesize that this could be because DPN patients respond differently to the different vibration intensities administered to the three intervention groups. The continuous vibration administered to the patients in group 3, might work well initially, but when patients get used to the continuous vibration they become desensitized to the vibratory stimulus, thus diminishing the effect of the intervention. Alternatively, in groups 1 and 2 the changing vibratory frequencies seemed more effective as the patients receiving this type of vibration took longer to get accustomed to the vibration pattern. This is further supported by the high satisfaction scores of individuals in intervention group 2. The high satisfaction scores and the high compliance rate observed among the participants of this study corroborated each other. Future studies will be needed to test the hypothesis regarding the varying response to a vibration stimulus. If true, the results could then be used to optimize and customize the vibrations delivered in FMV therapy to lie within a safe and effective range. The presence of outliers or unequal sample group sizes are other possible explanations for the lack of significance in the results from the ANOVA.

Due to uneven distribution and a small number of participants in the AT group, we did not find significant differences in the changes of those outcome measures between the AT and non-AT group. It was expected that the participants in the AT group would have worse balance and mobility at baseline as shown in Table 5. It was also encouraging to observe the clinically significant improvement of balance in the AT group (from 30.67 in average to 38.67), even if their balance was still worse than those in Non-AT group post intervention. In a future study we might be able to increase the dosage or provide a longer period of vibration to examine whether the balance for those using AT can be further improved. It was surprised to observe that the non-AT group improved more in TUG than the AT group did. This could have been because during the mobility test, none of those AT users had used their assistive devices, which might have affected their confidence in walking. But for TUG cognitive, both groups showed similar improvement, which indicated a reduced risk of falls. The AT group showed slightly worse pain at baseline compared to the non-AT group and more improvement, but it was not significant. Part of the reason for the non-significant improvement could be that our participants overall experienced less pain compared to those suffering painful DPN.

Two out of the 23 participants did not log their use of the device and four of the remaining 21 participants who were recruited during the COVID-19 pandemic logged between 24–45 sessions of FMV because their post-intervention (four-weeks from baseline) visit had to be rescheduled (The suggested intervention protocol was three sessions/week for four weeks for a total of 12 sessions). However, the device usage log obtained from all those 21 participants showed a 100% compliance rate. While most participants followed the suggested protocol diligently, a few participants logged four sessions some weeks and only two sessions other weeks, while still ensuring completion of at least 12 sessions prior to their post-intervention visit. Overall, some participants deviated from the intervention protocol slightly, but it was only observed among participants rescheduled due to the pandemic. The researchers also observed three participants who used the device more than suggested as they enjoyed using it. One potential problem with the compliance result is that usage was logged by the participants, which is subjective. We have been working on a new version of the technology which comes with an app to log usage of the device. Thus, in future studies, we will be able to monitor the FMV usage better and calculate the compliance.

The QUEST 2.0 satisfaction survey revealed that of the 22 participants who completed the survey, only three had scores under 30 and they were all dissatisfied with the straps to secure the device. One of the participants with a score of 24 was concerned about the lack of notable differences while using the device. This participant also admitted to mild memory impairment during the intervention, and failed to complete the device usage log. The other two participants with the low satisfaction scores had similar complaints and both disliked the charging cables and the need to charge the devices daily. One of those two, with the lowest satisfaction score (17), was recruited during the COVID-19 pandemic and complained about the lack of activity because of it. In addition, the participant’s vibration device malfunctioned.

The majority of the participants reported using the device at night while in a seated position. The feedback was mostly positive and the participants reported “loving” the size, weight, and the FMV device’s ease of use. The perceived benefits reported by participants included being more active, walking longer distances, and increased confidence in walking. One of the patients reported that she felt “more energetic and relaxed after using the device”, “had decreased back pain”, and was “able to shop for longer periods without fatigue”. Another patient reported he was “more active because of the device”, “slept better and felt better”. One of the common complaints observed was the insufficient strength of vibration. There was also consensus among all participants regarding the discomfort and difficulty in handling the straps that held the device in place when applying FMV. These findings demonstrate the promising nature of wearable FMV as an intervention for individuals with DPN. Because wearable FMV is ideal for in-home intervention, it can address the challenges of compliance when compared to interventions that are administered in a clinical setting such as WBV, spinal cord stimulation, and exercise training.

## 5. Study Limitations

A control or comparison group using traditional exercise intervention was not included, so investigators cannot exclude a placebo effect. This exploratory study did not include a control group because it is difficult to “blind” the participants, and they did not want to be in the no-vibration control group, especially after they experienced FMV during their first visits. The investigators have improved the FMV technology to be able to deliver no-effect sham vibrations (very low frequency with sound effect). Future study with better methodology, such as cross-over design and updated technology will enable comparisons on the effects of FMV between intervention and control groups. Nevertheless, the data collected will be helpful to effect size calculation, and this pilot protocol can guide future studies on the effectiveness and efficacy of FMV in individuals with DPN. The lack of a statistically significant improvement in balance raised concern about the reliability and validity of using the BBS to detect small changes in postural stability or balance in those individuals with DPN who possess a higher level of functional balance [38,40]. This study also raised concerns regarding the clinical assessment of DPN since SWMT was the only outcome used in this study that detected the presence or severity of DPN. We did not collect clinical parameters of diabetes severity such as HbA1C. These clinical parameters could have affected FMV’s influence on gait. In future studies, we will assess the medical record of participants and track their HbA1C both at baseline and post-intervention. Another confounder was the impact of the COVID-19 pandemic on this study. Due to restrictions on research activities as well as the safety of the participants, some post-intervention visits could not be scheduled according to the set protocol, and this resulted in some participants using the vibration device for longer than four weeks. However, during the interview we also learned from the participants that they had to cancel some of their clinic visits due to the pandemic but were able to apply the vibration therapy at home. This further supports the contention that a wearable FMV could be beneficial for those who have limited access to healthcare resources or for anyone during a situation like COVID-19. Although the post-hoc analyses revealed that the study was not sufficiently powered, this was a pilot study conducted to obtain the effect sizes to better power future studies.

## 6. Conclusions

This pilot study explored the benefits of wearable FMV on pain, mobility and sensation in individuals with DPN. The clinical implications of the findings of this study are still uncertain since the sample size was small and the effect sizes were low to moderate. Since the use of FMV as an intervention is associated with minimal risk, the findings of this study warrant further study with a larger sample size and more accurate outcome measures to determine the safety, effectiveness and efficacy of wearable FMV in individuals with DPN.

## Figures and Tables

**Figure 1 ijerph-18-02415-f001:**
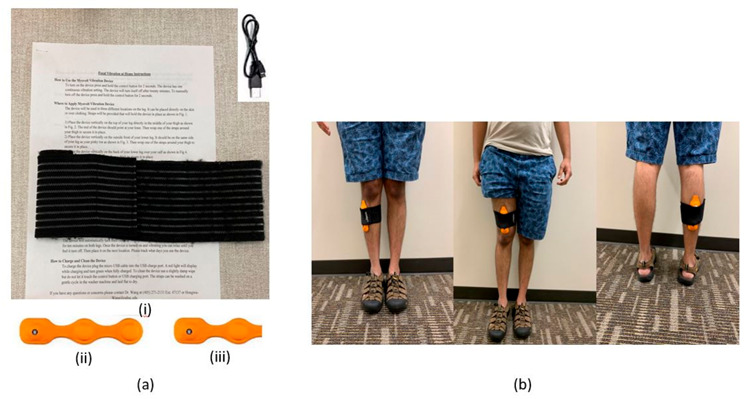
(**a**) (**i**) Charging cable, user manual, and strap for the modified Myovolt^TM^ device, (**ii**) Original Myovolt^TM^ vibration device with two vibration motors capable of multiple vibration intensities, (**iii**) Modified version of the Myovolt^TM^ with one vibration motor capable of only one vibration intensity; (**b**) **Left**: Modified Myovolt^TM^ attached to the tibialis anterior muscle, **Center**: Modified Myovolt^TM^ attached to the distal quadriceps muscle, **Right**: Modified Myovolt^TM^ attached to the belly of the gastrocnemius/soleus muscle.

**Figure 2 ijerph-18-02415-f002:**
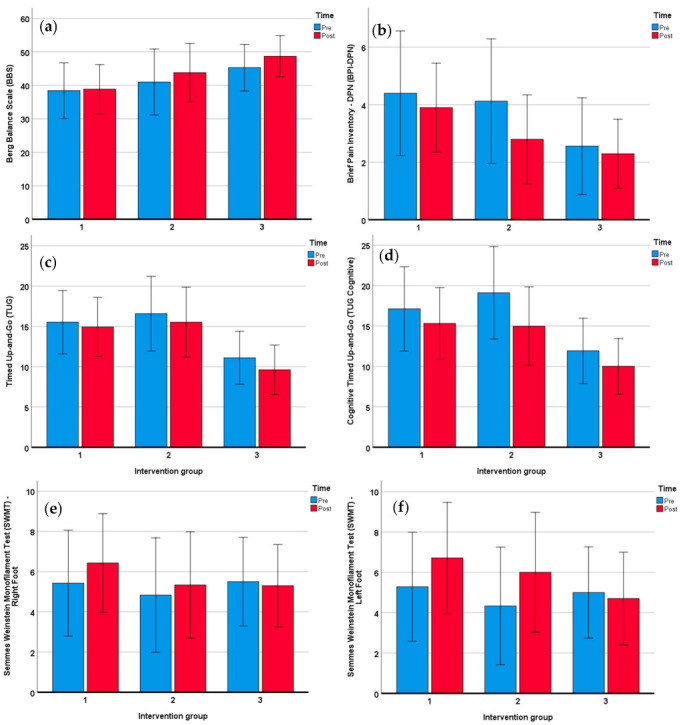
(**a**) Pre- and post-pain scores for each vibration group; (**b**) Pre- and post-balance scores for each vibration group; (**c**) Pre- and post-left foot sensation scores for each vibration group; (**d**) Pre- and post-right foot sensation scores for each vibration group; (**e**) Pre- and post-left TUG scores for each vibration group; (**f**) Pre- and post-TUG cognitive scores for each vibration group.

**Table 1 ijerph-18-02415-t001:** Patient demographic information ^#^.

Demographics/Group	All Participants	Intervention Group 1	Intervention Group 2	Intervention Group 3	F-Statistic	*p*-Value
N	23	7	6	10	-	-
Age (years) *	66.74 (10.76)	66.71 (13.35)	66.50 (5.75)	66.90 (12.08)	0.66	0.53
Weight (lb) *	219.00 (58.14)	194.14 (44.21)	240.33 (62.49)	223.60 (63.11)	1.08	0.36
Height (in) *	67.46 (3.58)	65.86 (3.90)	67.50 (4.28)	68.55 (2.77)	0.99	0.39
Body Mass Index (BMI) (lb/in2) *	33.72 (8.28)	31.19 (5.28)	37.16 (8.89)	33.42 (9.60)	0.84	0.45
Number of Years with Diabetes *	17.83 (8.41)	16.43 (5.56)	21.00 (13.31)	16.90 (6.69)	0.56	0.58
Sex (F/M)	14/9	5/2	3/3	6/4	-	-
Ethnicity						
Caucasian	21	6	6	9	-	-
African-American	1	0	0	1	-	-
Eurasian	1	1	0	0	-	-

F/M means Female/Male; Intervention group 1: Received a pulsed on-and-off vibration at 120 Hz; Intervention group 2: Received sinusoidal vibration ranging between 35 and 120 Hz; Intervention group 3: Received continuous vibration at a constant frequency of 120 Hz; * Indicates that values are represented as mean (standard deviation); F-statistic and p-value from one-way Analysis of Variance (ANOVA) shows no significant difference in the demographics between the different groups. ^#^ This table was published in our previous manuscript Rippetoe et al. [33].

**Table 2 ijerph-18-02415-t002:** Pain, balance, mobility, and sensation scores before and after a 4-week focal muscle vibration (FMV) intervention for all participants.

Outcome Measure and Time of Measurement		N	Mean (SD)	Mean Difference (Mean ± SD or Median)	95% Confidence Interval	*p*-Value	Effect Size
BPI-DPN	Pre	22	3.48 (2.56)	0.62 ± 1.40	(−0.004, 1.24)	0.051 ^^^	0.45
	Post		2.87 (1.85)				
BBS ^¥^	Pre	22	42.14 (10.48)	1.5	(−0.5, 5)	0.11	−0.24
	Post		44.45 (9.86)				
TUG	Pre	22	13.75 (5.34)	1.10 ± 2.30	(0.09, 2.12)	0.04 *	0.49
	Post		12.65 (5.25)				
TUG cognitive	Pre	21	15.12 (6.60)	2.41 ± 3.32	(0.90, 3.92)	0.003 *	0.73
	Post		12.71 (5.57)				
SWMT Right foot ^¥^	Pre	23	5.30 (3.19)	0	(−0.5, 1)	0.43	−0.12
	Post		5.65 (3.01)				
SWMT Left foot	Pre	23	4.91 (3.29)	−0.74 ± 2.12	(−1.65, 0.18)	0.12	−0.37
	Post		5.65 (3.45)				

^¥^ means data is not normally distributed; * means *p* < 0.05 and result is statistically significant; ^ means Trending towards significance; BPI-DPN is Brief Pain Inventory–Diabetic Peripheral Neuropathy; BBS is Berg Balance Scale; TUG is standard Timed Up-and-Go; TUG cognitive is the cognitive Timed Up-and-Go; SWMT is Semmes–Weinstein Monofilament Test; N is number of participants analyzed; BPI-DPN scores are measured out of 110 and the scores are scaled to a score between 0–10.

**Table 3 ijerph-18-02415-t003:** Item scores on BPI–DPN pre- and post-intervention in individuals with DPN.

BPI–DPN Subscales	N	Pre	Post	Mean/Median Difference	95% Confidence Interval	*p*-Value
Worst Pain ^¥^	21	4.48 (3.28)	4.05 (2.94)	0	(−1.5, 0.5)	0.53
Least Pain ^¥^	22	2.68 (2.98)	1.91 (1.57)	0	(−2, 0.5)	0.31
Average Pain	22	4.00 (2.29)	3.18 (2.26)	0.82 (1.30)	(0.24, 1.39)	0.007 *
Current Pain ^¥^	22	2.77 (2.76)	2.18 (1.65)	0	(−1.5, 0.5)	0.38
General Activity	22	3.14 (3.39)	2.23 (1.88)	−1	(−2, 0.5)	0.07 ^^^
Mood	22	2.82 (3.02)	3.05 (2.82)	0	(−1, 1)	0.63
Walking ability ^¥^	22	4.14 (3.20)	3.09 (1.97)	−1	(−2, 0)	0.03 *
Interference to Normal walking	22	3.68 (2.93)	3.14 (2.55)	0.55 (2.74)	(−0.67, 1.76)	0.36
Relationships ^¥^	22	1.95 (2.80)	1.86 (2.10)	0	(−0.5, 0.5)	0.95
Sleep ^¥^	23	4.61 (3.94)	4.00 (3.36)	−0.5	(−1.5, 0.5)	0.23
Enjoyment	22	3.82 (3.72)	3.27 (2.47)	−0.5	(−2, 1)	0.48

All values represented as mean (SD); ^¥^ means data is not normally distributed; * means *p* < 0.05 and the result is statistically significant; ^ means trending towards significance. The first 4 items in the BPI–DPN measure pain severity where Worst Pain is worst pain experienced in the last 24 h; Least Pain is the least pain experienced in the last 24 h; Average Pain is the average pain experienced in the last 24 h; and Current Pain is the pain experienced at the time of measurement. The other items of the BPI–DPN measure pain-related interference, where General Activity measures pain-related interference to general activity; Mood measures pain-related interference to mood; Walking Ability measures pain-related interference to walking ability; Interference to normal walking measures pain-related interference to normal walking; Relationships measures pain-related interference to relationships; Sleep measures pain-related interference to sleep; and Enjoyment measures pain-related interference to enjoyment.; Each item is scored between 0–10.

**Table 4 ijerph-18-02415-t004:** Baseline and post-intervention scores for balance, mobility, pain, and sensation for each intervention group.

Group/ Outcomes	N	BBS		BPI–DPN		TUG		TUG Cognitive		SWMT Left		SWMT Right	
		Pre	Post	Pre	Post	Pre	Post	Pre	Post	Pre	Post	Pre	Post
Intervention group 1	7	38.43 (13.01)	38.85 (12.38)	4.41 (2.24)	3.89 (1.21)	15.52 (6.11)	14.93 (5.35)	17.11 (6.92)	15.32 (6.27)	5.29 (3.25)	6.71 (3.15)	5.43 (3.36)	6.43 (3.05)
Intervention group 2	6	41.00 (11.96)	43.80 (11.65)	4.17 (3.31)	2.80 (1.90)	16.58 (5.25)	15.52 (6.18)	19.11 (7.91)	14.98 (6.72)	4.33 (3.83)	6.00 (3.79)	4.83 (3.54)	5.33 (3.50)
Intervention group 3	10	45.30 (7.57)	48.80 (4.37)	2.56 (2.12)	2.31 (2.02)	11.10 (3.85)	9.61 (3.06)	11.93 (4.44)	10.01 (3.40)	5.00 (3.29)	4.70 (3.53)	5.50 (3.21)	5.30 (2.91)

BPI–DPN is Brief Pain Inventory-Diabetic Peripheral Neuropathy; BBS is Berg Balance Scale; TUG is standard Timed Up-and-Go; TUG cognitive is the cognitive Timed Up-and-Go; SWMT is Semmes–Weinstein Monofilament Test; N is number of participants analyzed; BPI–DPN scores are measured out of 110 and the scores are scaled to a score between 0–10; All values are represented as mean (SD).

**Table 5 ijerph-18-02415-t005:** Baseline and Post-intervention scores for balance, mobility, pain, and sensation for AT and non-AT group.

Group/ Outcomes	N	BBS		BPI-DPN		TUG		TUG Cognitive		SWMT Left		SWMT Right	
		Pre	Post	Pre	Post	Pre	Post	Pre	Post	Pre	Post	Pre	Post
AT Group	7	30.67 (12.03)	38.67 (9.61)	3.87 (2.57)	3.04 (2.15)	17.46 (5.77)	17.05 (6.01)	19.83 (7.93)	17.61 (7.02)	5.14 (3.44)	5.14 (3.98)	5.57 (3.59)	5.29 (3.15)
Non-AT Group	16	46.44 (5.76)	46.69 (9.31)	3.33 (2.61)	2.80 (1.76)	12.36 (4.62)	10.99 (3.98)	13.89 (5.62)	10.75 (3.52)	4.81 (3.33)	5.88 (3.30)	5.19 (3.12)	5.81 (3.04)

## Data Availability

The data presented in this study are available on request from the corresponding author. The data are not publicly available due to ethical reasons and to protect the privacy of the participants.

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
