# Peer review of "Wearable Focal Muscle Vibration on Pain, Balance, Mobility, and Sensation in Individuals with Diabetic Peripheral Neuropathy: A Pilot Study"

_ijerph, 2021, doi:10.3390/ijerph18052415_

Round 1
Reviewer 1 Report
General Comments:
- This is an overall solid study with one major methodological limitation, there is no control group. I believe that authors have rightfully, therefore, referred to this as an “exploratory” study and have addressed this limitation directly in the discussion.
- At the editor’s discretion, a modification to the title, indicating “pilot”, may be in order.
Specific Comments:
- Abstract results should include values (mean ±SD), not just p-values
- Line 74, Contrast is made between WBV and Focal “its application can easily be kept within safe limits” provide citation
- Line 74 and 75, benefits of FMV – support/cite
- These are some generalized claims regarding FMV, but germane to your thesis. It is not clear that these claims are directly related to the outcomes of this study. Clarify, cite
- Line 99-100 – Stating that there were 3 groups labeled group 1, 2, and 3 is not helpful. What were the different groups?
- Line 127-130 – what was the rationale for these 3 different groups? Support for these different conditions? Do we know the intensity (amplitude) of the vibration stimulus?
- 35-120Hz is a very wide range, can we be more specific? If not, why not? Was this a limitation?
- Why was there no control group?
- Clarify why the modified motor was used instead of the original. Rationale?
- Line 181 and subsequent – Are “Mode 1 and 2” the same as Intervention groups 1 and 2? This should be clarified and consistent throughout
- Table 2: clarify that this is combined data
- Given the choice to separate groups (despite no rationale provided), do we know that it is reasonable to combine analysis across groups?
Author Response
General Comments:
- This is an overall solid study with one major methodological limitation, there is no control group. I believe that authors have rightfully, therefore, referred to this as an “exploratory” study and have addressed this limitation directly in the discussion.
Thank you for the suggestion. We acknowledged that lack of a control group is a limitation of this study and agreed that this is an exploratory study.
- At the editor’s discretion, a modification to the title, indicating “pilot”, may be in order.
Thank you. We added A pilot to Study to the title of the manuscript.
Specific Comments:
- Abstract results should include values (mean ±SD), not just p-values
Thank you for the suggestion. We added the mean ±SD in the abstract.
- Line 74, Contrast is made between WBV and Focal “its application can easily be kept within safe limits” provide citation
Thank you. The citation for the statement was added.
- Line 74 and 75, benefits of FMV – support/cite
- These are some generalized claims regarding FMV, but germane to your thesis. It is not clear that these claims are directly related to the outcomes of this study. Clarify, cite
Thank you for the suggestion. We agreed that those statements are general. There was no previous work on benefits of FMV on DPN but we believe that some of the benefits of FMV on people with neurological disorders might be applied to DPN. We added the citation to make it clear.
- Line 99-100 – Stating that there were 3 groups labeled group 1, 2, and 3 is not helpful. What were the different groups?
Thank you. We included the vibration delivered to each group to make it clear.
- Line 127-130 – what was the rationale for these 3 different groups? Support for these different conditions? Do we know the intensity (amplitude) of the vibration stimulus?
Thank you for the questions. These are great questions. We added the rationale for the 3 different groups. Our primary goal for this exploratory study is to examine whether FMV may work for DPN. Due to the wide vibrations were used in the literature, and the capacity of the wearable device we have that can provide three vibration modes, we decided to explore whether patients with DPN would respond differently to different vibration intensities. As for the intensities of the vibration stimulus, they were not reported by the manufacture. We did tested the peak-to-peak vibration amplitude for the three modes, and found out that the amplitudes are dependent with the frequencies, and they were between 4.1 to 5.3 g's for the three modes (the higher the frequency, the higher the intensity).
- 35-120Hz is a very wide range, can we be more specific? If not, why not? Was this a limitation?
Thank you for the comment. We intentionally picked this range as we want to see whether alternating the frequency (the intensity was altered at the same time as mentioned above). We can pick specific frequency in our future studies as the device can now deliver whatever frequency to be set by the clinician. But based on our findings, the change of the frequencies actually delivered better effect as it was difficult for the users to get used to the stimulation. We agreed with you this should be further studied.
- Why was there no control group?
Thank you for the question. We briefly explained this in the revised paper. One reason was we were not able to deliver sham vibration at the time of the study so we can not assign a control group with sham vibration. Another reason was that this is a pilot study and we did not have enough incentives to keep the participants in the control group. The participants refused to participate the study if they were assigned to control group, especially after they tried the device during their first visit. We may either consider a cross-over design in the future, or increase the incentives to keep a control group.
- Clarify why the modified motor was used instead of the original. Rationale?
Thank you for the question. We added the reason for the modified motor in the revised paper. The original device came with two vibration motors in each unit, and each unit has the three vibration modes. When we tested the actual device, we noticed that the two vibration motors interfere with each other so it will be difficult for us to quantify the vibration delivered. In addition, we did not want the participants to change the vibration mode by themselves, so we asked the manufacture to modify the device to one motor with one of the three modes each unit.
- Line 181 and subsequent – Are “Mode 1 and 2” the same as Intervention groups 1 and 2? This should be clarified and consistent throughout
Thank you for the suggestion. Yes, the Mode 1 and 2 are the same as Intervention groups 1 and 2. We changed all modes to Intervention groups in the revised paper.
- Table 2: clarify that this is combined data
- Given the choice to separate groups (despite no rationale provided), do we know that it is reasonable to combine analysis across groups?
Thank you for the comments. Yes Table 2 is the combined data. As we mentioned above, the primary objective of this pilot study is to examine whether FMV would benefit patients with DPN. The three vibration groups were also exploratory objectives based on our hypothesis that patients might be respond differently to the three different vibration intensities.
Reviewer 2 Report
People with diabetic peripheral neuropathy (DPN) experience lower quality of life caused by associated pain, loss of sensation and mobility impairments.
In this field, the present paper adds useful knowledge and insights. Rationale, methods, data and interpretations are well founded.
This study explored the benefits of 4-week in home wearable focal muscle vibration therapy on useful attributes of diabetic peripheral neuropathy (pain, balance, mobility, sensation). In itself, the subject is worthy of investigation in as much as alternative therapies for diabetic peripheral neuropathy are still being sought.
The rationale is clearly explained by the authors. The Introduction is clear and eloquent. Study design, criteria and methods are duly provided and explained. Procedures are well described. The statistical analysis is free from flaws. Results are clearly and logically presented. The Discussion highlights major findings and provides reasonable interpretations. Limitations are frankly acknowledged. Implications are clearly summarised.
Tables and Figures are necessary and well chosen.
Author Response
People with diabetic peripheral neuropathy (DPN) experience lower quality of life caused by associated pain, loss of sensation and mobility impairments.
Thank you for your feedback. We appreciate it.
In this field, the present paper adds useful knowledge and insights. Rationale, methods, data and interpretations are well founded.
Thank you for your feedback. We appreciate it.
This study explored the benefits of 4-week in home wearable focal muscle vibration therapy on useful attributes of diabetic peripheral neuropathy (pain, balance, mobility, sensation). In itself, the subject is worthy of investigation in as much as alternative therapies for diabetic peripheral neuropathy are still being sought.
Thank you for your feedback. We appreciate it.
The rationale is clearly explained by the authors. The Introduction is clear and eloquent. Study design, criteria and methods are duly provided and explained. Procedures are well described. The statistical analysis is free from flaws. Results are clearly and logically presented. The Discussion highlights major findings and provides reasonable interpretations. Limitations are frankly acknowledged. Implications are clearly summarised.
Thank you for your feedback. We appreciate it.
Tables and Figures are necessary and well chosen.
Thank you for your feedback. We appreciate it.
Reviewer 3 Report
This is a straight forward paper about home based vibrations therapy for DPN. The big weakness is the lack of control group or other comparative therapy, which the authors fully note. The manuscript is well written and the authors do a nice job comparing their outcomes with the relevant literature.
The authors should mention that other outcomes reported on this study have been previously published (Rippetoe et al., 2020) in the paragraph on subjects. The number in each group, ages, etc are all the same, therefore this is the same group previously reported on.
As in that study, this group should be a mixed bag of mobility, yet the inclusion criteria states ambulate independently. Which is it? Please provide some information on how the group that used assistive devices improved relative to the non device users of the group. As it stands, there is little information about these subjects using assistive devices.
Figure 2 y-axis is low resolution and does not need 2 significant digits. Revision suggested.
Author Response
This is a straight forward paper about home-based vibrations therapy for DPN. The big weakness is the lack of control group or other comparative therapy, which the authors fully note. The manuscript is well written and the authors do a nice job comparing their outcomes with the relevant literature.
Thank you for the feedback. We acknowledge that the lack of a control group is a limitation and agree that this is an exploratory study. We have also modified the title to include “A Pilot Study” in it to make it more appropriate. We also further explained why we did not have a control group in this exploratory study and our future plan to make sure a control group be included.
The authors should mention that other outcomes reported on this study have been previously published (Rippetoe et al., 2020) in the paragraph on subjects. The number in each group, ages, etc are all the same, therefore this is the same group previously reported on.
Thank you for the feedback. We made a note of this in Table 1 in the revised paper.
As in that study, this group should be a mixed bag of mobility, yet the inclusion criteria states ambulate independently. Which is it? Please provide some information on how the group that used assistive devices improved relative to the non-device users of the group. As it stands, there is little information about these subjects using assistive devices.
Thank you for the feedback. The group of participants did have varying mobility, but they were all able to ambulate independently. We have edited the methods section now to include the definition of “ambulate independently” to make it more clear to the readers. “Ambulate independently” is defined as “ability to ambulate without supervision or physical assistance from another person. Assistive devices, orthoses, and prostheses are allowed”.
We appreciated your suggestion on checking the AT and non-AT group and we did find some interesting results which were included in the Table 5 of the revised paper. We also added a paragraph in the discussion session to discuss the findings and we believe this could be a future study to further investigate. Thank you for your insights.
Figure 2 y-axis is low resolution and does not need 2 significant digits. Revision suggested.
Thank you for the feedback. The y-axis of figure 2 has been edited.
Reviewer 4 Report
The research team should wait to publish the results. They should increase the sample of each of the groupsAuthor Response
The research team should wait to publish the results. They should increase the sample of each of the groups.
Thank you for the comments. We agreed with you that our results could be strengthened with more participants in each group. While we are still trying our best to continue the recruitment, there are couple reasons for this submission:
- We found based on the current data, our primary hypothesis that FMV may benefit patients with DPN was supported, at least partially based on the improvements on pain and mobility. The current data also supports us to further study why the balance was improved by some of the participants but not all. It also supports us to include more clinical measures of DPN in our future studies. We are preparing a NIH R21 based on the current pilot results to examine the mechanism of FMV on DPN with a focus on the nerve motor and sensory responses to FMV.
- Since the COIVD pandemic, we’ve had challenge to recruit new participants. Even for the few new participants, their visits were constrained by the modified research policy by the university and local government.
- This is a pilot study supported by an exploratory fund within University of Oklahoma Health Sciences Center. We are on no cost extension now and we are about the wrap up the study even though our original recruitment goal of each group with 12 participants are not met.
Round 2
Reviewer 4 Report
The number of volunteers is still small. There is no calculation of the sample size. ANOVA application criteria are not explainedAuthor Response
The number of volunteers is still small. There is no calculation of the sample size. ANOVA application criteria are not explained.
Thank you for those great comments. We acknowledged that the number of volunteers is still small, especially for the intervention group1 and 2.
We had a sample size estimation when planned the study. We added the below sample size calculation to the session 2.2 of the manuscript: "Sample size. Because no previous research has been conducted about the effect of focal vibration in patients with DPN, we performed power calculations using minimal clinically important differences (MCID) when this information was available, as well as reasonable estimates of expected change. We based our calculations on our primary outcomes TUG, pain and BBS. With an alpha = 0.05, power of 0.8, and utilizing a paired t-test, we will need eight subjects per group to detect within subject changes."
In addition, we performed a post-hoc power calculation and added into the result session: "A post-hoc power analysis was conducted on the BPI-DPN, TUG, and TUG cognitive scores as they were the three outcomes with a moderate effect size greater than 0.4. This post-hoc analysis revealed that, based on the BPI-DPN scores (effect size = 0.45), TUG scores (effect size = 0.49), and TUG cognitive scores (effect size = 0.73), the study was powered at 52%, 59%, and 88% respectively."
As for ANOVA application criteria, we added those statements in the session 2.6 "The assumptions of residuals in both within and between subjects be normally distributed, homogeneity of variances or homoscedasticity using the Box’s M statistic, and Mauchly's test of sphericity for the ANOVA were examined and used to report the appropriate ANOVA results.".